# Therapeutic Advances Propelled by Deciphering Tumor Biology and Immunology—Highlights of the 8th Heidelberg Myeloma Workshop

**DOI:** 10.3390/cancers13164135

**Published:** 2021-08-18

**Authors:** Raphael Lutz, Mirco Friedrich, Marc Steffen Raab, Niels Weinhold, Hartmut Goldschmidt

**Affiliations:** 1Department of Hematology, Oncology and Rheumatology, University Hospital Heidelberg, 69120 Heidelberg, Germany; raphael.lutz@med.uni-heidelberg.de (R.L.); mirco.friedrich@med.uni-heidelberg.de (M.F.); marc.raab@med.uni-heidelberg.de (M.S.R.); niels.weinhold@med.uni-heidelberg.de (N.W.); 2Division of Stem Cells and Cancer, German Cancer Research Center (DKFZ), 69120 Heidelberg, Germany; 3Clinical Cooperation Unit Molecular Hematology/Oncology, German Cancer Research Center (DKFZ), 69120 Heidelberg, Germany; 4National Center for Tumor Diseases, 69120 Heidelberg, Germany

**Keywords:** multiple myeloma, immunotherapy, measurable residual disease

## Abstract

**Simple Summary:**

The 8th Heidelberg Myeloma Workshop was held on 16–17 April 2021 at the University Hospital Heidelberg, Germany. The main topics of the meeting were diagnostics and prognostic factors of early-phase multiple myeloma (MM), the role of immunotherapy, as well as the biology and genomics of MM. This manuscript reports on recent advances in MM research and points out future directions.

**Abstract:**

The diagnostics and treatment of newly diagnosed and relapsed MM are continuously evolving. While advances in the field of (single cell) genetic analysis now allow for characterization of the disease at an unprecedented resolution, immunotherapeutic approaches and MRD testing are at the forefront of the current clinical trial landscape. Here, we discuss research progress aimed at gaining a better understanding of this heterogenous disease entity, presented at the 8th Heidelberg Myeloma Workshop. We address the questions of whether biology can guide treatment decisions in MM and how assessment for measurable residual disease can help physicians in clinical decision-making. Finally, we summarize current developments in immunotherapeutic approaches that promise improved patient outcomes for MM patients. Besides summarizing key developments in MM research, we highlight perspectives given by key opinion leaders in the field.

## 1. Can Biology Guide Treatment Decisions in MM?

In the past few decades, research has illustrated that MM is not a uniform disease entity. Despite recent advances in therapeutic options, a deep understanding of MM disease biology is urgently needed in order to personalize therapy strategies and to help to drive clinical decision-making regarding when to treat MM patients.

Gareth Morgan from the Perlmutter Cancer Center in New York, USA emphasized the importance of disease prevention in his keynote lecture on MM disease biology as symptomatic MM cases are preceded by a precursor disease developing over 20–30 years [1]. Along these lines, Niccolò Bolli from the University of Milan, Italy highlighted a recently published study that compared the genomic features of stable and progressing early MM stages independently of the definition of MGUS or smoldering myeloma [2]. Using whole-genome sequencing, the authors were able to show lower mutational activity, fewer copy number aberrations and hardly any complex rearrangement in stable monoclonal gammopathies, whereas the progressing early MM stages showed a genomic profile reminiscent of symptomatic MM. Current risk stratification according to the International Myeloma Working Group (IMWG) criteria for smoldering myeloma implements clinical laboratory parameters as well as cytogenetic alterations but are not covering the full spectrum of MM disease biology [3]. To date, clinical decision-making on starting a first-line therapy is still based on the presence of the Slim CRAB criteria [4]. Bolli and Morgan both argued that early treatment of progressing precursor disease states has the potential to prevent irreversible end organ damage. They proposed a paradigm shift in which the definition of MGUS and smoldering myeloma based on quantitative disease activity parameters should be replaced by a characterization based on MM biology. Thus, the long-term goal is to define targeted sequencing panels and further biomarkers that could predict progression to symptomatic MM in order to optimize disease prevention strategies.

According to Morgan, risk-stratified therapy in first-line treatment using the ISS or R-ISS is largely applicable to patient groups but lacks specificity for individual patients. Morgan thereby envisions that future optimized individual risk stratification should implement comprehensive approaches including copy number aberrations, gene expression patterns and assessment for complex structural variants [5,6].

Risk stratification based on MM biology becomes even more challenging when incorporating the spatial heterogeneity of MM. Niels Weinhold from the University Hospital Heidelberg, Germany presented a multi-region sequencing study showing that different subclones grow at the sides of nodular plasma cell infiltrations, so-called focal regions [7]. Hereby, the authors found a positive correlation between the size of a focal region and the presence of side-specific clones. Based on these studies, Morgan, Weinhold and Leo Rasche from the University Hospital in Würzburg, Germany argued for an extended model of MM evolution with two phases. The first phase is characterized by selective sweeps of tumor clones in the early disease stage. In later disease stages, regional evolution takes place, leading to a broad range of spatial heterogeneity within one patient. Morgan compared the intra-patient heterogeneity of MM disease with a tree containing different branches of regional subclonal evolution. He points out that targeting a subclone will lead to the outgrowth of another subclone, highlighting the need for therapy strategies that target the roots of the tree instead of the branches. A prime example of a druggable early event in MM disease evolution is the translocation t(11;14). The survival and growth of MM cells harboring t(11;14) are strongly dependent on the expression of high levels of the antiapoptotic protein BCL-2 [8]. Thus, Venetoclax is capable of inducing cell death in such MM cells by selectively inhibiting BCL-2 [9,10].

Morgan stresses that personalizing MM therapy is already a reality in MM treatment as therapies are chosen based on the risk stratification of a patient and with regard to individual comorbidities and the patient’s general condition. Future personalized therapy strategies should rely on multi-omics approaches that define distinct molecular subgroups. How such molecular patterns can lead to targeted therapy strategies was illustrated by the pivotal role of RAS/MAPK signaling in MM cell survival [11,12]. Targeting this pathway by MEK inhibition or BRAF inhibitors has shown promising results in early clinical studies [13].

The disease evolution model becomes even more complex in relapsing patients. Weinhold presented a recently published single-cell expansion model suggesting that relapse might be driven by single tumor cells [14]. Such single tumor cells have been tracked postmortem based on the acquisition of unique mutations in each cell occurring during Melphalan High-Dose treatment. Weinhold showed multiple examples indicating that tumor evolution during treatment is a very complex and not a uniform process. A longitudinal study including functional imaging and multi-region sequencing of MM patients during treatment suggests that clonal selection occurs during therapy, but more importantly, clonal evolution is ongoing and leads to highly resistant subclones. These results again underline the importance of early intervention and highly effective therapy strategies that incorporate basic knowledge on resistance mechanisms.

Along these lines, the role of cereblon as therapeutic target in MM has been discussed by Jan Krönke from the Charité Hospital in Berlin. Cereblon (CRBN) is the primary target by which immunomodulatory drugs (IMiDs) such as thalidomide and its derivatives, lenalidomide and pomalidomide, mediate their anti-tumor effects. Besides the approved drugs, further IMids with higher potency, such as avadomide, iberdomide and CC-92480, are currently under clinical investigation and are even promising in the case of resistance to approved IMids [15,16,17].

Together, emerging technical advances including multi-omics approaches combined with imaging techniques have deepened our understanding of disease evolution in early precursor states as well as in symptomatic and refractory myeloma. Translating this basic knowledge into the clinic has great potential to further improving clinical outcomes but it remains a major challenge to account for the inter- and intra-patient heterogeneity of MM patients with personalized therapy strategies. This encourages further research on large, comprehensive datasets incorporating innovative techniques. As the disease complexity increases over time, Morgan concludes his keynote lecture by pointing out the power of early prevention in MM precursor stages, which should be a major focus of future MM research.

## 2. Ready for Guiding Therapeutic Decisions? MRD, Functional Imaging and Future Approaches

Currently, minimal residual disease in MM patients is detected by Next-Generation Flow (NGF) or Next-Generation Sequencing (NGS)-based analysis of bone marrow (BM) samples after therapy. Stefanie Huhn from the University Hospital in Heidelberg, Germany, however, emphasizes that this only reflects a snapshot of the disease state as disease activity is only evaluated at a specific time point and at a single location. To acknowledge the intra-patient spatial heterogeneity of MM patients in remission, the term minimal residual disease has been replaced by measurable residual disease (MRD).

Due to intensive research efforts in the past decade, MRD is well established in MM and represents the most powerful prognostic factor according to Bruno Paiva from the University of Nevarra in Pamplona, Spain. Regardless of the used MRD detection method, MRD negativity predicts improved survival in both standard and high-risk patients, justifying the role of MRD status as a primary endpoint in clinical studies [18]. Hereby, reaching MRD negativity at a sensitivity level of 10^−5^ has been shown to be beneficial to predict patient outcomes. The MRD status has proven to be a useful endpoint after induction therapy even in elderly patients and, more recently, in MM patients with high-risk cytogenetic aberrations [19]. Paiva points out that high-risk FISH patients with persistent MRD should be candidates for innovative therapies already in early therapy lines due to their poor prognosis. Thereby, MRD positivity is a very reliable parameter and defines patients with suboptimal response that might need a new line of therapy.

In contrast, measurement of MRD negativity is a less reliable parameter as it requires high-quality BM samples with a low degree of hemodilution. In addition, there is a need for continuous monitoring of patients to capture the state of sustained MRD negativity over 12 months, which defines a subgroup with ultra-low risk of progression, as shown in the CASTOR and POLLUX clinical trials for patients with relapsed or refractory MM [20].

Functional imaging via PET/CT and DW-MRI is an additional approach accounting for intra-patient spatial heterogeneity in MRD assessment. Elena Zamagni from the University of Bologna, Italy introduced the 18F-FDG-PET/CT to assess for metabolic activity in the BM in order to monitor treatment responses [21]. Zamagni and colleagues recently found that a complete metabolic response—defined by FDG uptake in focal lesions and in bone marrow lower compared to liver tissue (Deauville Score of liver = 4)—is an independent predictor for progression-free survival as well as overall survival [22].

Zamagni, Huhn and Leo Rasche from the University Hospital in Würzburg, Germany agreed that MRD assessment by NGF and NGS as well as functional imaging approaches are complimentary approaches leading to an optimized assessment of MRD.

Finally, Huhn also presented minimally invasive techniques as an emerging field in MRD research. Tracking of Circulating Tumor Cells (CTCs) as well as tracking of circulating DNA fragments of MM cells in peripheral blood has additional potential to capture MRD otherwise undetectable by conventional BM biopsies. Detection of CTCs in PB has been shown to be highly predictive for a positive MRD status in BM, with a specificity of 95–100%, but has limited sensitivity to detect MRD in comparison to BM biopsies [23]. If the sensitivity could be increased, measuring MRD via CTC might have the potential to avoid sequential BM punctures. Furthermore, future clinical research needs to address whether MRD detection via minimally invasive techniques informing about the spatial heterogeneity of MM subclones translates into better prediction of patient outcomes than a bone marrow test.

Only recently, a mass spectrometry-based method termed MALDI-MS has been introduced as an alternative approach for the classical electrophoresis and immunofixation performed in clinical routine [24,25]. In addition, this method could also be beneficial in MRD detection, which should be addressed in future studies.

Taken together, the above observations indicate that MRD assessment is well-established as an endpoint in clinical trials. Hereby, it has the potential to recalibrate patients’ risk after treatment and could be used as a starting point for future patient-individualized concepts. However, data from ongoing randomized clinical trials are still pending to ultimately guide therapy in clinical practice.

## 3. Current and Future Treatment Standards: Next Chapter Immunotherapy?

Although the prognosis of patients suffering from MM—even after relapse—has steadily improved in recent years, the therapy success of the most modern next-generation substances merely results in chronic disease. Ever since the approval of adoptive cell therapy for the treatment of acute lymphatic B cell leukemia (ALL) and diffuse large B cell lymphoma (DLBCL), cellular immunotherapies have been a main focus in hematology and oncology.

Several potential targets have been identified as suitable for anti-myeloma Chimeric Antigen Receptor (CAR) T cell therapy, with B cell maturation antigen (BCMA) being the most developed [26,27]. BCMA is a cell surface receptor of the tumor necrosis factor (TNF) receptor superfamily that is preferentially expressed on mature B cells including plasma cells and is important for B cell development, proliferation and survival [28]. However, BCMA expression may be variable since it can be cleaved by γ-secretase, leading to shedding from the cell surface. The toxicity of currently available cellular therapies as well as other immunotherapies, such as bispecific T cell engagers, often includes immunological syndromes, such as cytokine release syndrome (CRS), of varying intensity, as well as immune effector cell-associated neurotoxicity syndrome (ICANS) or prolonged cytopenia. An overview of clinical trials with published (interim) results investigating CAR-T cell therapy in MM is presented in Table 1.

Michael Hudecek from the University Hospital Würzburg presented other novel targets in early development phases, including signaling lymphocytic activation molecule family member 7 (SLAMF7), which is expressed on a variety of lymphocytes, including B and T cells, NK cells and plasma cells. SLAMF7 is more widely known as the target of the monoclonal antibody elotuzumab. The development of CAR-T cells directed against SLAMF7 may be more challenging, however, due to its expression on T cell subsets, which may lead to fratricide [29]. G protein-coupled receptor class C group 5 member D (GPRC5D) is another promising target that is physiologically expressed in hair follicles, but also highly expressed on plasma cells [30]. The bispecific antibody talquetamab targeting GPRC5D has already shown promising response rates in heavily pretreated patients [31].

More advanced data have been reported by Nikhil C. Munshi of the Dana-Farber Cancer Center in Boston, MA, on the CAR-T product Idecabtagen Vicleucel (ide-cel, bb2121). In the phase 1 study of ide-cel, 33 patients have been treated with escalating doses of 50–800 × 10^6^ total modified T cells. The ORR was 85%, with a 45% CR rate and a PFS of 11.8 months for patients who received ≥150 × 10^6^ CAR-T cells. Toxicity was comparatively low, with grade 3/4 CRS of 6% and ICANS of 3% [32,33]. In the phase 2 KarMMa trial using ide-cel, 128 patients were treated. ORR was 73%, with a CR rate of 33% and median PFS of 8.8 months. This pivotal study has since led to the FDA approval of ide-cel, with EMA approval currently pending. Grade 3/4 CRS was 5% and ICANS 3% [34]. In the ongoing KarMMa-2 study, ide-cel is being evaluated in patients with early relapse after first-line therapy or patients with remission status < VGPR after auto-SCT. The phase 3 study KarMMa-3 is evaluating ide-cel vs. standard-of-care regimens in RRMM. The disadvantages of autologous CAR-T cells are the long production times and reduced T cell fitness due to the heavy pretreatment of patients in current clinical trials. This may be overcome in part by preemptive T cell harvesting early during the course of the disease. Off-the-shelf or allogeneic CAR-T cells may be another alternative that was intensively discussed during the meeting.

Marc S. Raab from the University Hospital Heidelberg, Germany, introduced the design and function of bispecific T cell engagers (Figure 1).

Raab summarized recent developments in the spectrum of bispecific T cell engagers, of which several are in clinical trials targeting various antigens of myeloma cells, including CD38, GPRC5D and BCMA [35,36,37,38,39] (Table 2).

Teclistamab is one of several bispecific T cell engagers targeting BCMA, with the rationale of binding both BCMA and CD3 on the surfaces of T cells and thereby activating them so that they induce cell apoptosis and direct lysis of malignant plasma cells that are in proximity. A phase 1 study in patients who were triple- or penta-class refractory showed impressive ORRs in the range of 70% to 80% at the randomized phase 2 dose. Raab adds that one important drawback of these bispecific T cell engagers is their short half-life, which potentially affects efficacy. Moreover, the specific mechanisms of the induced immune response by bispecifics are currently not well understood. Another candidate targeting BCMA, AMG-701, therefore features an integrated Fc domain to extend the biological half-life. In a dose-escalation study of triple-class refractory patients, ORR for AMG-701 was shown to be dose-dependent. Participants receiving the lowest dose of 0.015–1.6 mg achieved an ORR of 4%, whereas the ORR of participants receiving the highest evaluated dose of 9 mg was 83% [36].

These preliminary results of novel bispecific T cell engagers were deemed to be very promising, though important open questions remain, such as long-term survival due to the currently short follow-up phase, potential combinatorial regimes or immune-related toxicity.

Sagar Lonial of Emory University, GA, further outlined the pleiotropic therapeutical options of BCMA, adding that antibody–drug conjugates (ADCs) might play an important role in anti-CD38-refractory patients. Belantamab mafodotin (Belamaf) is the first ADC in MM to be approved by EMA as a monotherapy in August 2020. Professor Lonial demonstrated data of the phase II DREAMM-2 study that led to the approval of Belamaf in MM patients that were refractory against proteasome inhibitors, immunomodulators and anti-CD38 antibodies [45]. The ORR in patients receiving a dose of 2.5 mg/kg was 31% as the median PFS has not been reached yet. A worst-case analysis estimated the PFS at 9 months. The most important adverse effect is keratopathy, which occurred in 27% of patients receiving Belamaf. Encouragingly, most patients that suffered from Belamaf-associated keratopathy experienced a full recovery of their vision upon dose modification or cessation of treatment. Professor Lonial therefore stressed that close interdisciplinary cooperation between the treating hematologist and ophtalmologist is essential [46].

In summary, novel immunotherapies including bispecific T cell engagers, ADCs and CAR-T cell therapies promise high response rates, even in heavily pretreated patients. Implementing such a therapy in MM treatment requires technical expertise and clinical experience in managing immunotherapy-specific toxicity. Widely used, these novel immunotherapy regimens could revolutionize myeloma therapy in the near future. However, some important questions remain unanswered, such as immunological and neurological long-term effects as well as the appropriate patient selection.

## 4. Conclusions

In recent years, translational research based on novel methods has led to a deeper understanding of the heterogenous disease entity Multiple Myeloma and propelled therapeutic advances. Here, we report on current views on key questions in MM research, which were discussed at the 8th Heidelberg MM Workshop. Understanding MM biology from precursor stages to refractory disease, as well as the optimized detection of measurable residual disease (MRD) after therapy, will be key to guide future personalized treatment decisions. Immunotherapeutic approaches including CAR-T cells, bispecific antibodies as well as antibody–drug conjugates (ADC) are clearly moving to the forefront in the clinical trial landscape. Thus, this report also focuses on recent developments in immunotherapy presented by key opinion leaders in the field and gives a comprehensive overview of current clinical trials. In the future, immunotherapeutic approaches will be also implemented in larger clinical trials on newly diagnosed MM patients and are expected to revolutionize MM therapy.

## Figures and Tables

**Figure 1 cancers-13-04135-f001:**
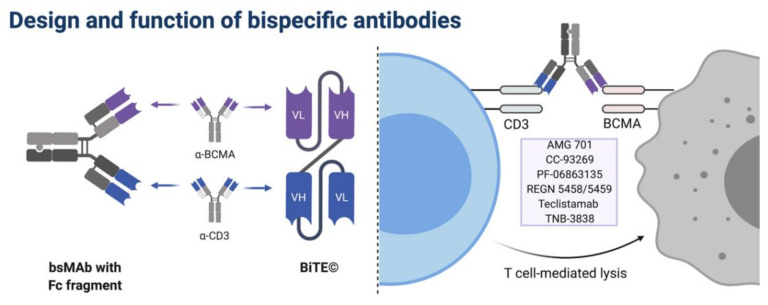
Design and function of bispecific antibodies. Bispecific antibodies connect the antigen-recognizing fragments of two different monoclonal antibodies and lead to binding of CD3^+^ T cells to cells expressing the target of interest (e.g., BCMA for MM cells). Currently investigated anti-BMCA bispecific antibodies are listed on the right. BCMA, B-cell maturation antigen.

**Table 1 cancers-13-04135-t001:** Overview of clinical trials with published (interim) results investigating CAR-T cell therapy in multiple myeloma.

Drug Study ID	Target	Estimated Sample Size	Co-Stimulation	Response Rate	Safety Profile
HRAIN BCMA-CART NCT03093168	BCMA	49	4-1BB + EGFRt	sCR/CR (45%)/VGPR (18%)	CRS grade 1–2 (12%), grade ≥ 3 (6%)
BCMA-CART ChiCTR-18000017404	BCMA	33/32	4-1BB	sCR/CR (66%)/VGPR (22%)	CRS grade 1–2 (52%), grade ≥ 3 (48%)
LCAR-B38M NCT03548207	BCMA	113	4-1BB	sCR/CR (67%)/VGPR (26%)	CRS grade 1–2 (90%), grade ≥ 3 (5%) neurotoxicity (10%)
LCAR-B38M NCT03090659 Phase-2	BCMA	57	4-1BB	sCR/CR (73%)/VGPR (4%)	CRS grade 1–2 (82%), grade ≥ 3 (7%) neurotoxicity (2%)
LCAR-B38M NCT03090659 Phase-1	BCMA	17	4-1BB	sCR/CR (82%)/VGPR (6%)	CRS grade 1–2 (59%), grade ≥ 3 (41%)
P-BCMA-101 NCT03288493	BCMA	23/19	4-1BB + rimiducidSS	sCR/CR + VGPR (26%)	CRS grade 1–2 (9%) neurotoxicity (4%)
FHVH33 NCT03602612	BCMA	15/42	4-1BB	sCR/CR (20%)/VGPR (7%)	CRS grade 1–2 (87%), grade ≥ 3 (7%) neurotoxicity (27%)
MCARH171 NCT03070327	BCMA	10/11	4-1BB + EGFRt	VGPR (45%)	CRS grade 1–2 (40%), grade ≥ 3 (20%) neurotoxicity (10%)
SZ-MM-CART01 NCT03196414	BCMA	29/28	CD28/OX40	sCR/CR (54%)/VGPR (4%)	CRS grade 1–2 (66%), grade ≥ 3 (34%) neurotoxicity (3%)
SZ-MM-CART02 NCT03455972	BCMA	32	CD28/OX40	sCR/CR (72%)/VGPR (ND)	CRS grade 1–2 (97%), grade ≥ 3 (3%)
FCARH143 + GSI NCT03502577	BCMA	10	4-1BB + EGFRt	sCR/CR (30%)/VGPR (50%)	CRS grade 1–2 (60%), grade ≥ 3 (40%) neurotoxicity (60%)
FCARH143 NCT03338972	BCMA	11	4-1BB	sCR/CR (55%)/VGPR (36%)	CRS grade 1–2 (91%) neurotoxicity (9%)
CART-BCMA + CTL119 NCT03549442	BCMA CD19	16	4-1BB	sCR/CR (19%)/VGPR (25%)	CRS grade 1–2 (88%)
CT103A ChiCTR-1800018137	BCMA	18	4-1BB	sCR/CR (67%)/VGPR (17%)	CRS grade 1–2 (72%), grade ≥ 3 (22%)
JCARH125 NCT03430011	BCMA	44	4-1BB	sCR/CR (27%)/VGPR (20%)	CRS grade 1–2 (70%), grade ≥ 3 (9%) neurotoxicity (25%)
CT053 NCT00302403 NCT03380039 NCT03716856	BCMA	24	4-1BB	sCR/CR (79%)/VGPR (4%)	CRS grade 1–2 (63%) neurotoxicity (8%)
CART-BCMA UPenn NCT02546167	BCMA	25	4-1BB	sCR/CR (8%)/VGPR (20%)	CRS grade 1–2 (56%), grade ≥ 3 (32%) neurotoxicity (32%)
BM38 CAR ChiCTR-1800018143	BCMA	22	4-1BB	sCR/CR (55%)/VGPR (9%)	CRS grade 1–2 (68%), grade ≥ 3 (23%)
NCI BCMA CAR-T NCT02215967	BCMA	16	CD28	sCR/CR (13%)/VGPR (50%)	CRS grade 1–2 (56%), grade ≥ 3 (38%) neurotoxicity (6%)
NCI BCMA-CAR-T NCT02215967	BCMA	10	CD28	VGPR (10%)	CRS grade 1–2 (30%)
BCMA-CART ChiCTR-OPC16009113	BCMA	28	CD28 + 4-1BB	sCR/CR (61%)/VGPR (4%)	CRS grade ≥ 3 (14%)
bb21217 NCT03274219	BCMA	38	4-1BB	sCR/CR (13%)/VGPR (34%)	CRS grade 1–2 (61%), grade ≥ 3 (5%) neurotoxicity (24%)
bb2121 NCT02658929 Phase-1	BCMA	43/39	4-1BB	sCR/CR (44%)/VGPR (23%)	CRS grade 1–2 (58%), grade ≥ 3 (5%) neurotoxicity (33%)
bb2121 NCT03361748 Phase-2	BCMA	149	4-1BB	sCR/CR (33%)/VGPR (26%)	CRS grade 1–2 (84%), grade ≥ 3 (5%) neurotoxicity (18%)
CD19 & BCMA-CAR-T ChiCTR-OIC17011272	CD19 BCMA	21	4-1BB	sCR/CR (57%)/VGPR (24%)	CRS grade 1–2 (86%), grade ≥ 3 (5%) neurotoxicity (10%)

BCMA, B cell maturation antigen; CAR, chimeric antigen receptor; CR, complete response; CRS, cytokine release syndrome; EGFRt, truncated epidermal growth factor receptor; ND, not disclosed; PR, partial response; sCR, stringent complete response; ChiCTR, Chinese Clinical Trial Registry; VGPR, very good partial response. Data as of 23 January 2021.

**Table 2 cancers-13-04135-t002:** Overview of clinical trials investigating bispecific antibodies.

Drug Study ID	Target	Phase	Estimated Sample Size	Response Rate	Safety Profile	References
AMG 420 (BI 836909) NCT02514239	BCMA × CD3	I	120	ORR: 31% CR: 8/42	CRS 38% grade 3/4: 29%	[40]
AMG 424 NCT03445663	CD38 × CD3	I	20	study terminated prematurely		
AMG 701 NCT03287908	BCMA × CD3	I	135	ORR: 23.2%	CRS 61% grade 3/4: 8%	[36]
Blinatumomab NCT03173430	CD19 × CD3	I	20	study terminated prematurely		
CC-93269 NCT03486067	BCMA × CD3	I	19	ORR: 40% CR: 5/30	CRS 77% grade 3/4: 9% grade 5: 1 death	[35]
Cevostamab NCT03275103	FcRH5 × CD3	I	80	-	CRS 74.5% grade 3/4: 2%	[41]
GBR 1342 NCT03309111	CD38 × CD3	I	125	results expected for 2021		
PF-06863135 NCT03269126	BCMA × CD3	I	80	CR: 2/18	CRS 61% grade 3/4: 67%	[37]
REGN 5458 NCT03761108	BCMA × CD3	I/II	56	ORR: 35.8%	CRS 88.2% grade 3/4: 0%	[38]
REGN 5459 NCT04083534	BCMA × CD3	I/II	56	results expected for 2023		
Talquetamab NCT03399799	GPRC5D × CD3	I	185	i.v.: ORR 67% s.c.: ORR 66%	i.v.+ s.c.: CRS 47% i.v. grade 3/4: 8% s.c. grade 3/4: 0	[42]
Teclistamab NCT03145181	BCMA × CD3	I	120	ORR: 63.8% CR: 9/120	i.v. CRS 53% s.c. 50% grade 3/4: 0%	[43]
TNB-3838 NCT03933735	BCMA × CD3	I	72	ORR: 37% CR: 3/38	21% grade 3/4: 0%	[44]

ORR, overall response rate; CR, complete response; CRS, cytokine release syndrome; data as of 23 January 2021.

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
