# Peer review of "Therapeutic Advances Propelled by Deciphering Tumor Biology and Immunology—Highlights of the 8th Heidelberg Myeloma Workshop"

_cancers, 2021, doi:10.3390/cancers13164135_

Round 1

Reviewer 1 Report

The article is a nice piece of work. The authors give a nice comprehension about the treatment of multiple myeloma with immunotherapy and MM
 relapsed/refractory. Many compounds are listed, their biological/medicinal effects are presented in a very detailed way. I'd like to recommend to publish this manuscript. No other suggestions.

Author Response

Dear Reviewer,

 We thank you for recommending our conference report entitled “Therapeutic advances propelled by deciphering tumor biology and immunology – Highlights of the 8th Heidelberg Myeloma Workshop” for publication in Cancers Journal.

Reviewer 2 Report

This report on the highlights of the 8th Heidelberg Myeloma Workshop by Lutz and colleagues summarizes important recent advances in understanding disease biology, measuring minimal residual disease and employing immune targets for treatment. It discusses limitations of current knowledge and provides thoughts from key opinion leaders on future directions. It provides a valuable concise summary for people interested in myeloma who were not able to attend the conference.

I have no major concerns but noted minor errors and have listed comments that the authors may consider in pursuit of further improvements of their manuscript:

  1. Title: consider replacing “immunity” with “immunology” since this fits the content of the review better. Immunity suggests responses of the patients’ immune system to their myeloma. In the manuscript this is not described but development of immunologic targets for myeloma treatment is described.
  2. Line 95-96: It think the authors mean to say “effective”, not “efficient” which is a poorly characterized term for description of therapeutic modalities.
  3. Line 150-159: The description of minimally invasive blood MRD assessments would benefit from a comparison of their sensitivities. To the less well versed reader, highlighting only the positive predictive value of eg detection of circulating tumor cells may give the false impression that these tests may also currently have value in assessing for MRD negativity which is obviously the main goal of MRD assessment. In line 153 to 154 it sounds like bone marrow biopsies can already be avoided. I would suggest to modify this sentence to convey that bone marrow biopsies may be avoided in the future if the sensitivity of minimally invasive blood tests can be increased and if the added value to inform about spatially heterogeneous clones translates into better prediction of outcome than a bone marrow test.
  4. Chapter 3 introductory paragraph and Table 1: Consider introducing cytokine release syndrome (CRS) and immune effector cell-associated neurotoxicity syndrome (ICANS) in the first paragraph as main expected adverse events. In table 1 consider adding ICANS as a separate item in the safety profile for each study to convey which studies even reported the incidence and you might want to consider adding a subtitle to table 1 that conveys whether studies have used the same criteria for assessing CRS and ICANS.
  5. Line 198: ICANS is listed without being spelled out anywhere in the manuscript. Please see previous comment.
  6. Line 200: I realize this review has a data cut-off of Jan 2021 but you might want to add here or elsewhere below that pivotal KarMMa study has in the meanwhile led to FDA approval of this BCMA CAR-T therapeutic.
  7. Table 2: In the safety profile column at several locations “grade” lacks an “e” and is misspelled as “Grad”. In addition, the row describing the Teclistamab study the ORR is erroneously listed as 25% when it was reported at ASH 2020 as 63.8%.
  8. Line 226 – 228: This sentence requires clarification. “Raab adds…short half-life…potentially affects efficacy; even though the specific mechanisms of an induced adaptive immunity are not well understood.” It should be explained what is meant with adaptive immunity here and how this would affect half-life or efficacy. Is suspect this was an error with copy-pasting, at least it reads like that.
  9. Line 234-236: This sentence is not very well formulated. Suggest considering to use “questions” instead of “issues” and to be clearer about what is meant with “overall survival”. Is the intention to communicate that follow up is relatively short on these studies or that randomized controlled trials should compare them to other compounds or CAR-Ts to assess their value at extending overall survival?
  10. Line 252: Mentioning cure achieved in single cases of solid tumors with novel immunotherapies appears misplaced in the summary of chapter 3 and it is also not clear to which of the three listed immunotherapies this pertains. I don’t think this needs to be mentioned at all but if the authors feel it would be valuable to include, I would recommend to mention this in the introduction to chapter 3 and to be specific about which immunotherapies are meant that induced cure in solid tumor patients.  
